# T2I-Scorer: Quantitative Evaluation on Text-to-Image Generation via Fine-Tuned Large Multi-Modal Models

## ABSTRACT

Text-to-image (T2I) generation is a pivotal and core interest within the realm of AI content generation. Amid the swift advancements of both open-source (such as Stable Diffusion) and proprietary (for example, DALLE, Midjourney) T2I models, there is a notable absence of a comprehensive and robust quantitative framework for evaluating their output quality. Traditional methods of quality assessment overlook the textual prompts when judging images; meanwhile, the advent of large Multi-Modal models (LMMs) introduces the capability to incorporate text prompts in evaluations, yet the challenge of fine-tuning these models for precise T2I quality assessment remains unresolved. In our study, we introduce the **T2I-Scorer**, a novel two-stage training methodology aimed at fine-tuning LMMs for T2I evaluation. For the first stage, we collect 397K GPT-4V-labeled question-answer pairs related to T2I evaluation. Termed as **T2I-ITD**, the pseudo-labeled dataset is analyzed and examined by human, and used for instruction tuning to improve the LMM's low-level quality perception. The first stage model, **T2I-Scorer-IT**, has reached superior accuracy on T2I evaluation than all kinds of existing T2I metrics under zero-shot settings. For the second stage, we define an explicit multi-task training scheme to further align the LMM with human opinion scores, and the fine-tuned **T2I-Scorer** can reach state-of-the-art accuracy on both *image quality* and *image-text alignment* perspectives with significant improvements. We anticipate the proposed metrics can serve as a reliable metric to gauge the ability of T2I generation models in the future. We will make code, data, and weights publicly available.

## CCS CONCEPTS

• **Computing methodologies → Computer vision tasks**.

## KEYWORDS

Text-to-Image Generation, Evaluation, Large Multi-Modal Models

## 1 INTRODUCTION

Over recent years, text-to-image (T2I) generation has progressed swiftly, effectively transforming the way we interact with digital content and bridging the gap between linguistic creativity and visual representation. Represented by Stable-Diffusion [11, 46], DALLE [45], or MidJourney [17], state-of-the-art T2I models can not

Permission to make digital or hard copies of all or part of this work for personal or classroom use is granted without fee provided that copies are not made or distributed for profit or commercial advantage and that copies bear this notice and the full citation on the first page. Copyrights for components of this work owned by others than the author(s) must be honored. Abstracting with credit is permitted. To copy otherwise, or republish, to post on servers or to redistribute to lists, requires prior specific permission and/or a fee. Request permissions from permissions@acm.org.
*ACM MM, 2024, Melbourne, Australia*
© 2024 Copyright held by the owner/author(s). Publication rights licensed to ACM.
ACM ISBN 978-x-xxxx-xxxx-x/YY/MM
https://doi.org/10.1145/nnnnnnn.nnnnnnn

only generate high-quality and aesthetic images, but also generate images that follow specific text depictions (*i.e.* prompts).

Nevertheless, there still lacks a reliable objective system to evaluate the quality of T2I generation. On the one hand, image quality assessment (IQA) methods [3, 19, 74] only take the generated images as inputs, without consideration of their alignment with the text prompts. On the other hand, vision-language similarity-based metrics [29, 43, 67, 68] are usually not enough sensitive to traditional low-level quality issues (*color, clarity, brightness, etc*). Moreover, models based on CLIP [43] have proven to be inferior in understanding sentence-level complex semantics, making them also sub-optimal to sufficiently evaluate image-text alignment of T2I generation. Given these limitations of existing metrics on evaluating T2I generation, the majority of recent T2I generation models still use human studies to evaluate their abilities. The community is in need of a more comprehensive metric to evaluate T2I generation.

In this study, we propose a better T2I evaluator, **T2I-Scorer**, via discovering through the pivotal question:

*What abilities should a more reliable T2I evaluator possess?*

First, **can understand text prompts comprehensively,** which is inalienable on evaluating image-text alignment (Fig. 1(a)). While CLIP-based models typically fall short on it, recently emerging large Multi-Modal models (LMMs), as represented by GPT-4V [39] and Gemini [14], as well as excellent open-source counterparts [9, 30, 31, 70], have shown stronger ability on understanding more comprehensive text inputs or instructions [32, 72]. Though with fundamental visual and text understanding abilities, existing studies find zero-shot LMM evaluations [6, 22, 33, 65] only weakly correlate with human opinions. Henceforth, for the proposed **T2I-Scorer**, we propose to *fine-tune* an LMM for a better quantitative T2I evaluator.

Second, **can effectively perceive both high-level (*content*) and low-level (*quality*) visual attributes,** which is important for better fidelity assessment [26] on T2I generation. While open-source LMMs can generally perceive well on high-level objects and themes of images, several benchmark studies have pointed out that they still have a significant gap with GPT-4V in terms of low-level visual perception, especially while comparing multiple images [61, 64, 66, 79]. To improve open-source LMMs, we employ GPT-4V to collect 397K low-level-related question-answering pairs on 20K T2I-generated images, denoted as **T2I-ITD** (*Text-to-Image Instructional Tuning Dataset*). The **T2I-ITD** includes not only questions on single images, *e.g. How is the clarity of the image?*, but also on image pairs, *e.g. Which image is brighter?*, to better improve LMMs on quality-related perception. The **T2I-ITD** dataset is used for the first stage instruction training, yielding the **T2I-Scorer-IT**, an LMM capable of answering open-vocabulary questions on T2I-generated images. Additionally, as the answers in T2I-ITD are designed to be convertible to numerical levels (*e.g.* 1,2,3), the **T2I-Scorer-IT** can also provide quantitative evaluations on T2I generation, which is proven more accurate than existing metrics.

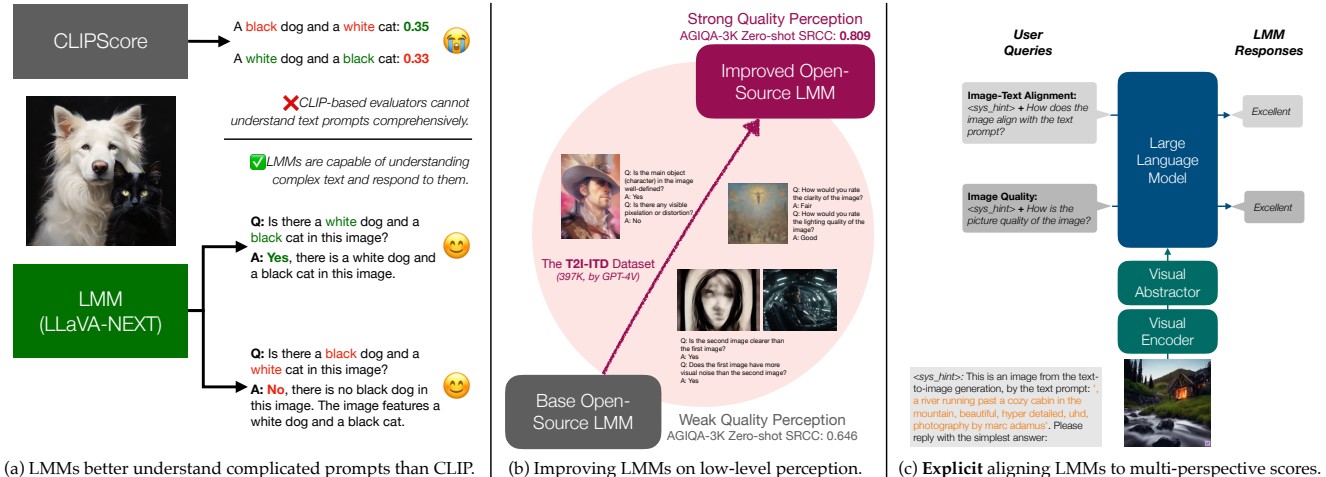

**Figure 1: The methodology of T2I-Scorer: (a) For comprehensive understanding of text prompts, we choose to fine-tune an LMM instead of CLIP as backbone structure. (b) For enhanced low-level visual perception, we collect the T2I-ITD dataset, with 397K question-answering pairs from GPT-4V. (c) For awareness of the perspectives for T2I evaluation (*alignment/quality*), we introduce an explicit multi-tasking syllabus with different question-answer templates for different perspectives.**

Third, **can be explicitly aware of different evaluation criteria for different perspectives.** While most existing image or video quality assessment approaches [12, 19, 56, 58, 75] are based on implicitly regressed scores, T2I evaluation has two different perspectives: *alignment/fidelity*, which might not be distinguished effectively under implicit regression. To avoid this problem, during the stage 2 fine-tuning on human opinion datasets, we convert the human opinion scores from two different perspectives into two different question templates that explicitly query the corresponding perspectives, *e.g. How does the image align with the text prompt?* for *alignment*. Then, we convert the continuous scores to ITU-standard [1] 5-point likert scales as answers.

In summary, we propose the **T2I-Scorer**, the first LMM-based scorer for T2I generation, with contributions as follows:

- We collect the **T2I-ITD**, a large-scale pseudo-label question-answering dataset for T2I evaluation. With 397K question-answer pairs on 20K T2I-generated images from GPT-4V, the question-answering dataset improves existing open-source LMMs on low-level quality-related perception.
- With **T2I-ITD**, we fine-tune an open-source LMM (mPLUG-Owl2) into **T2I-Scorer-IT**. The **T2I-Scorer-IT** can not only answer to quality-related questions, but also effectively provide quantitative scores. **T2I-Scorer-IT** reaches state-of-the-art accuracy on zero-sho T2I evaluation settings.
- We further fine-tune **T2I-Scorer-IT** under an explicit multi-tasking syllabus to further align with human opinions on *image-text alignment* and *image quality* perspectives. The fine-tuned **T2I-Scorer** achieves state-of-the-art performance.

## 2 RELATED WORKS

### 2.1 Text-to-Image (T2I) Generation

Recent advancement of Text-to-Image (T2I) generation has been pre-dominated by diffusion models, which have shown better image quality and image-text alignment than GAN-based generators [49].

As pioneer by latent diffusion [46], a typical diffusion-based T2I generation model [45, 48] consists of a VAE [20] encoder/decoder to convert images to/from latent spaces, a U-Net (or a transformer for most recent studies [11, 41]) to denoise latent inputs, and a CLIP or T5-XXL [44] text encoder to allow text control on the synthesized images. In short, state-of-the-art T2I generation models [8, 10, 17, 34, 42] have shown excellent abilities to synthesize images given a wide range of text prompts. Nevertheless, the abilities of current T2I generators still vary a lot under different text prompts and face failure cases (either low image quality or poor prompt following), calling for a robust and accurate objective metric for T2I generation.

### 2.2 Large Multi-Modal Models (LMMs)

Large Multi-Modal models (LMMs) are a novel type of Multi-Modality foundation models that generate text responses from visual and text inputs. As represented by LLaVA [31], InstructBLIP [7], mPLUG-Owl [69], LMMs typically include a visual encoder (such as CLIP [43], DINO [40], or SAM [21]), a text-only large language model (LLM) (*e.g* LLaMA [52]), and a connector between the two parts, which could be either a shallow multi-layer perception (MLP) or a deeper cross-attention abstractor [29, 69]. LMMs not only show superior ability on visual question answering benchmarks [2, 15, 35], but also show exciting abilities on understanding and following diverse user instructions [13, 31, 32, 72], and even seamlessly dialog with human about the visual inputs. In this work, we utilize the strong ability of LMMs to serve as backbones for the proposed T2I-Scorer.

### 2.3 Image Quality Assessment (IQA)

*IQA Methods.* Image quality assessment (IQA) is a traditional domain for multi-media studies. Traditional IQA methods, such as SSIM [55], NIQE [37], and ILNIQE [73] typically rely on statistical features to evaluate the quality of images and prove their effectiveness on artificial distortions (compression artifacts , white noise, gaussian blur, *etc*) [24, 25]. Nevertheless, these methods usually

do not work well on in-the-wild quality evaluation scenarios, as they are not aware of any visual semantics. On the contrary, deep-learning-based IQA methods [19, 27, 51, 74, 75] usually achieve better performance on these in-the-wild datasets. Among them, some methods [53, 59, 76] have adopted CLIP, by fine-tuning the softmax-pooled cosine similarity between images and text-defined levels (*e.g. good, poor*) as the quality score for images.

*IQA on T2I Images.* Recently, with the advancement of T2I generation, several subjective studies have collected IQA datasets [5, 26, 54, 77] for T2I-generated images. There are also several weakly-labeled human opinion databases [67, 68] for T2I-generated images. Typically, the T2I-IQA task includes an **image-text alignment** (or prompt alignment) perspective, and a pure **image quality** perspective. Most recently, Li *et al.* have collected AIGIQA-20K, a large-scale database containing 20K images from 15 different T2I generation methods. We utilize the 20K images from the database to generate the **T2I-ITD** dataset, for the first stage training of **T2I-Scorer**.

*LMMs for IQA.* Given the strong ability of LMMs, several pioneer studies have investigated LMMs for IQA. Several benchmarks [60, 66, 79, 81] have proved decent zero-shot quality assessment as well as related perception abilities, which is still notably behind GPT-4V and has plenty of room for improvement. Given this insight, several studies [18, 62, 71] have collected question-answering datasets to improve quality perception and evaluation abilities of LMMs, including not only single-image settings but also multi-image comparative settings [64]. Despite training LMMs for direct text outputs, a most recent study, Q-Align [63], also proposes a human-alike strategy for LMMs to output precise quantitative scores, reaching state-of-the-arts on traditional IQA datasets. Inspired from these explorations, we design a two-stage training scheme to fine-tune an LMM into the proposed **T2I-Scorer**.

## 3 THE T2I-ITD DATASET

In this section, we elaborate on the *Text-to-Image Instructional Tuning Dataset* (**T2I-ITD**). We first introduce the image preparation (Sec. 3.1), and then discuss prompts for GPT-4V to generate quality-related question-answering data (Sec. 3.2) on both single images and image pairs, resulting in 397K question-answer pairs. Finally, we analyze the reliability of the GPT-4V-annotations (Sec. 3.3).

### 3.1 Image Preparation

We include the images from the recently-released database with 20K images by *Li et al.*. The images are generated from 15 different T2I models [4, 8, 10, 16, 17, 34, 42, 45, 47, 48, 50], generated via diverse hyper-parameter configurations such as iterations, Classifier Free Guidance (CFG), resolution, aspect ratios. It excludes outdated Generative Adversarial Network (GAN) [49] and Auto-Regressive (AR) models. The prompts used to synthesize these images are sourced from real user inputs of the AIGC community, selected from DiffusionDB, undergoing filtering (*removing not-suitable-for-workplace prompts & prompts with special characters*) to ensure content diversity and appropriateness, resulting in 20K prompts for model inputs (*each image is with different prompt*), final yielding a large-scale and diverse database for the following GPT-4V pseudo labeling process.

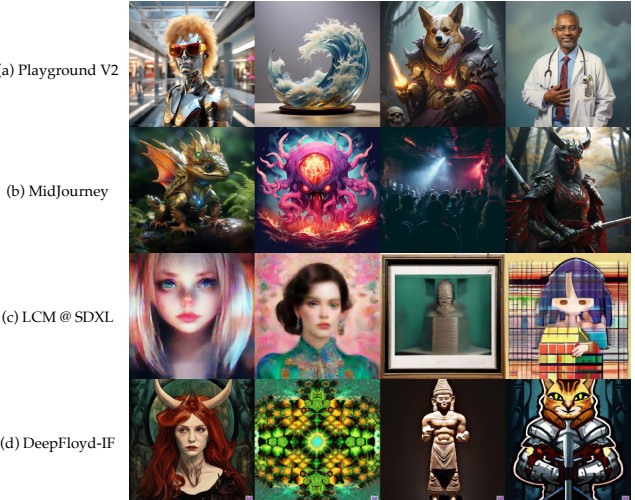

(a) Playground V2

(b) MidJourney

(c) LCM @ SDXL

(d) DeepFloyd-IF

**Figure 2: Images prepared for the T2I-ITD, synthesized via 15 T2I generation models on 20K real-world text prompts.**

### 3.2 "Labelling" the T2I-ITD with GPT-4V

*3.2.1 General Schemes.* Being used for the first training stage for the quantitative T2I-Scorer, the core principle of the T2I-ITD dataset is to make LMMs provide *easy to quantify* outputs. Henceforth, the answers for all question-answering data are designed to be distinguishable from the first word: each answer is either limited to one single word (*e.g.* Yes/No), or has more than one words but the first word is different for all possible answers (*e.g.* First image/Second image/Tie). This has allowed us to directly convert the logits for candidates in the *first token*, and further convert them into quantitative scores [60, 78]. Despite that, given existing observations [28, 57] that image contents notably affect both subjective and objective quality evaluations, we include the text prompts into the instructions for LMM fine-tuning. The instruction format for T2I-ITD is as follows, explicitly asking for the simplest answer:

**Single Images:** `This is an image from the text-to-image generation, by the text prompt: <prompt>. Please reply with the simplest answer: <question>`
**Image Pairs:** `This is a pair of images from text-to-image generation. The prompt for the first image is <prompt1>, and the prompt for the second image is <prompt2>. Please reply with the simplest answer: <question>`

We introduce the details of different question-answering subsets for single images (Sec. 3.2.2) and pairs (Sec. 3.2.3) as follows.

*3.2.2 Single Image Question-Answering.* The single image question-answering setting is the most common setting for LMMs. For single images, to facilitate the task for quantitative scoring, we collect two types of questions where answers could be converted into numerical values: *Yes-or-No* questions and *How* questions, as follows:

*Yes-or-No* questions contain binary judgements related to quality, *e.g. Is the lighting of the image good?*, with answer limited to [Yes, No]. The answers could map to numerical values as follows:

$$f_{YN} : \{Yes, No\} \rightarrow \{0, 1\}, \text{ where } f(Yes) = 1, f(No) = 0 \quad (1)$$

Being more 'fine-grained' than *Yes-or-No* questions, *How* questions contain multi-level evaluation related to quality, *e.g. How is the*

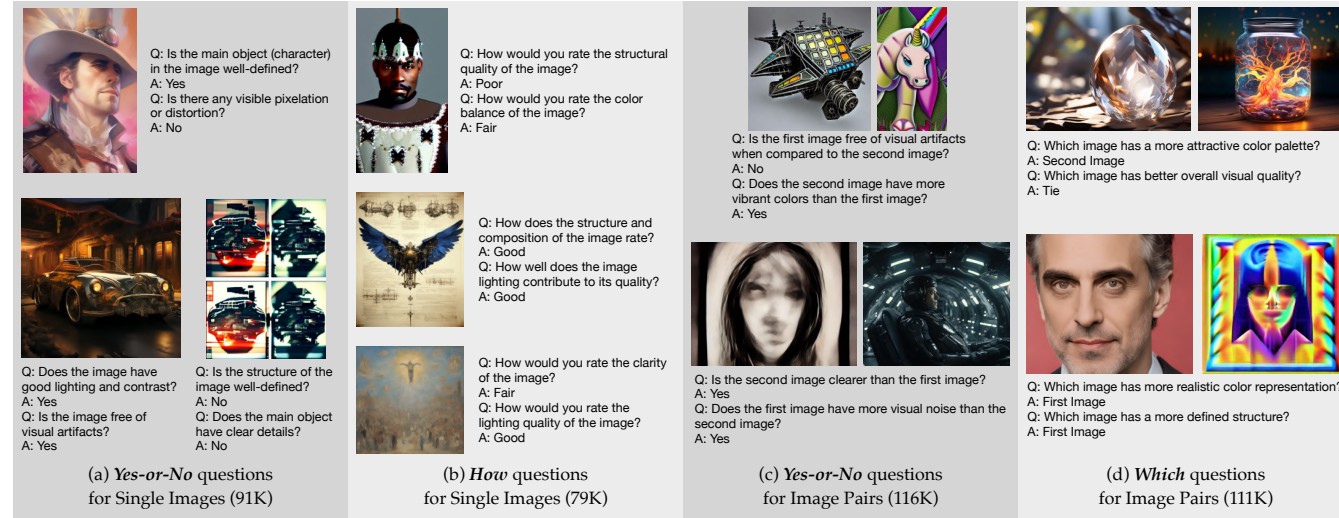

**Figure 3: Examples from 397K GPT-4V-generated question-answering pairs in the T2I-ITD dataset.**

*composition of the image?* To avoid free-form synonyms (*e.g. fair, average, medium*), we fix the answers to be one among [Good, Fair, Poor]. The answers could also map to numerical values:

$$f_{\text{How}} : \{\text{Good}, \text{Fair}, \text{Poor}\} \rightarrow \{0, 1, 2\},$$

where $f_{\text{How}}(\text{Good}) = 2, f_{\text{How}}(\text{Fair}) = 1, f_{\text{How}}(\text{Poor}) = 0$ (2)

The examples of GPT-4V generated pairs for *Yes-No* and *How* questions on single images are illustrated in Fig. 3 (a) and (b).

*3.2.3 Pairwise Question Answering.* Despite single image settings, many recent studies [66, 79] have reported GPT-4V's stronger quality-related perception ability on **image pairs** than single images. Henceforth, to better teach open-source LMMs, we similarly collect two types of question-answering data on pairwise visual inputs, including the *Yes-or-No* questions, which are generally the same with their counterparts for single images, only difference as comparative targets, *e.g. Does the second image appear to have better clarity?*, and an unique type of questions, the **Which** questions.

The **Which** questions raise a query related to quality, *e.g. Which image has better texture details?*, and the answer could only be chosen from **First image/Second image/Tie**. Similar to *How* questions, the answers of **Which** questions can also be converted into *relative* numerical scores (from perspective of first image):

$$f_{\text{Which}} : \{\text{First}, \text{Tie}, \text{Second}\} \rightarrow \{0, 1, 2\},$$

where $f_{\text{Which}}(\text{First}) = 2, f_{\text{Which}}(\text{Tie}) = 1, f_{\text{Which}}(\text{Second}) = 0$ (3)

The examples of GPT-4V generated pairs for *Yes-No* and *Which* questions on image pairs are illustrated in Fig. 3 (c) and (d).

## 3.3 Analysis on the T2I-ITD

As the **T2I-ITD** is labeled by GPT-4V instead of human annotators, it is important to analyze its composition and reliability before using it for training. For image composition, after GPT-4V labelling, we deliberately remove all data including images synthesized by DALLE-2 or DALLE-3 (5% of all images), which is reserved to examine the cross-model generalization of T2I-Scorer-IT. We further

discuss the composition of questions, the proportions of answers, and the reliability of the collected data as follows.

*3.3.1 Composition of Questions.* The 397K question-answer pairs are generally evenly composed of four question types. They include 97K *Yes-or-No* and 79K *How* questions on single images, as well as 116K *Yes-or-No* and 111K *Which* questions on image pairs. About the quality-related concerns, we use Wordcloud [38] to compute the frequencies of quality-related words in questions, and group the synonyms (*e.g. clarity, clear, clearly*) by GPT-4. Among all questions, the *clarity* dimension has the highest frequency (16.8%), followed by *lighting* (14.2%), *color* (10.3%), *composition* (6.7%), and *artifacts* (4.7%), with similar proportions to human quality descriptions in Q-Pathway [62]. Despite questions about whole images, there are also 7.4% questions asking about the main object, and 5.1% about the background, contributing to local in-context low-level perception [28, 60]. In general, these GPT-4V-raised questions have widely covered a rich variety of quality-related concerns.

*3.3.2 Proportions of Answers.* While the questions are evenly distributed with covering a rich variety of quality-related concerns, we observe that the answers are pretty "biased" towards positive responses (*i.e.*, Yes and Good). For *Yes-or-No* questions, the answer Yes makes up **69.3%** of all answers; for *How* questions, Good even shares **81.1%** among all answers. Nevertheless, the answers on *Which* questions are pretty balanced, with 44.1% for Second Image, 41.5% for First Image, and 14.4% for Tie. Therefore, such positive "bias" might come from the overall high quality of input T2I-generated images. To better understand the origin of the bias, we examine the data reliability of **T2I-ITD**, discussed as follows.

*3.3.3 Data Reliability.* To analyze the reliability of the **T2I-ITD**, we randomly sample 1500 questions from the whole database and ask 7 human experts to examine its correctness. Without seeing the GPT-4V answers, human experts independently choose one among all available answers for the question, and then we use a majority voting to determine the human label ($L_{\text{human}}$) of each

sample. For question types with three choices (*i.e. How&Which*), if no choices get > 3 votes, $L_{\text{human}}$ for this case will be regarded as '*Divergent*'; for *Yes-or-No* questions, a 4:3 vote is considered '*Divergent*'. Denoting GPT-4V answer as $L_{\text{GPT-4V}}$, the results of human evaluations on different question types are listed in Tab. 1, which primarily validates the reliability of the **T2I-ITD** dataset.

**Table 1: A human-involved sample analysis of GPT-4V generated answers on four question types. Samples with consistent human answers and GPT-4V answers are labeled in** gray **.**

**(a) Yes-or-No (Single Images)**

| $L_{\text{human}}$ | $L_{\text{GPT-4V}}$ | |
|---|---|---|
| | Yes | No |
| Yes | 203 (72%) | 7 (9%) |
| No | 27 (10%) | 60 (76%) |
| *Divergent* | 51 (18%) | 12 (15%) |
| Total | 281 | 79 |

**(b) How (Single Images)**

| $L_{\text{human}}$ | $L_{\text{GPT-4V}}$ | | |
|---|---|---|---|
| | Poor | Fair | Good |
| Poor | 24 (77%) | 2 (8%) | 11 (5%) |
| Fair | 3 (10%) | 18 (72%) | 21 (9%) |
| Good | 1 (3%) | 1 (4%) | 160 (70%) |
| *Divergent* | 3 (10%) | 4 (16%) | 36 (16%) |
| Total | 31 | 25 | 228 |

**(c) Yes-or-No (Image Pairs)**

| $L_{\text{human}}$ | $L_{\text{GPT-4V}}$ | |
|---|---|---|
| | Yes | No |
| Yes | 205 (75%) | 16 (10%) |
| No | 24 (9%) | 127 (76%) |
| Divergent | 45 (16%) | 24 (14%) |
| Total | 274 | 167 |

**(d) Which (Image Pairs)**

| $L_{\text{human}}$ | $L_{\text{GPT-4V}}$ | | |
|---|---|---|---|
| | 1st Img | Tie | 2nd Img |
| 1st Img | 139 (77%) | 5 (8%) | 8 (5%) |
| Tie | 13 (7%) | 48 (77%) | 11 (6%) |
| 2nd Img | 7 (4%) | 4 (6%) | 136 (79%) |
| *Divergent* | 21 (12%) | 5 (8%) | 18 (10%) |
| Total | 180 | 62 | 173 |

Despite general reliability, the human examination also comes with several conclusions: **(1)** Positive answers (Yes and Good) are only **slightly less accurate** than others, suggesting that the distribution of $L_{\text{GPT-4V}}$ mainly comes from overall acceptable quality of T2I-generated images; **(2)** GPT-4V answers on **image pairs** are slightly more accurate than single images, aligning with existing observations [79]. **(3)** Ignoring the human-'*Divergent*' samples, the GPT-4V answers can reach >**85%** agreement rate with human, proving them sufficient to serve as training data for the first stage.

## 4 THE T2I-SCORER

In this section, we discuss the structure and training of T2I-Scorer-IT and T2I-Scorer (Fig. 4), a family of LMMs with improved ability on T2I evaluation. For the choice of LMM, we adopt the mPLUG-Owl2 [70] structure. The LMM (denoted as **M**) includes a CLIP-ViT-L14 [43] visual encoder $\mathbf{E}_v$ with 304M parameters, a visual abstractor $\hat{\mathbf{E}}_v$ with 82M parameters, and a LLaMA2-7B [52] LLM (denoted as **L**). Denote input images as $\mathcal{I}$, previous text tokens as $\{t^0, \ldots, t^{N-1}\}$, the LMM autoregressively predict the logits for the $N$-th text token ($O^{N-1}$) as follows:

$$\mathcal{H}_v = \hat{\mathbf{E}}_v(\mathbf{E}_v(\mathcal{I})),$$
$$O^{N-1} = \mathbf{M}(\mathcal{I}, \{t^0, \ldots, t^{N-1}\}) \quad (4)$$
$$= \mathbf{L}(\mathcal{H}_v \oplus \mathbf{E}_t(\{t^0, \ldots, t_{\text{out}}^{N-1}\}))$$

where $\mathbf{E}_t$ is the text embedding layer, and $\mathcal{H}_v$ is the visual embedding, concatenated ($\oplus$) with $\mathbf{E}_t$ outputs and fed to LLMs.

The training scheme is conducted in two stages. In the first stage, we utilize the collected **T2I-ITD** to fine-tune the mPLUG-Owl2 into the **T2I-Scorer-IT** (Sec. 4.1), which is not only able to provide

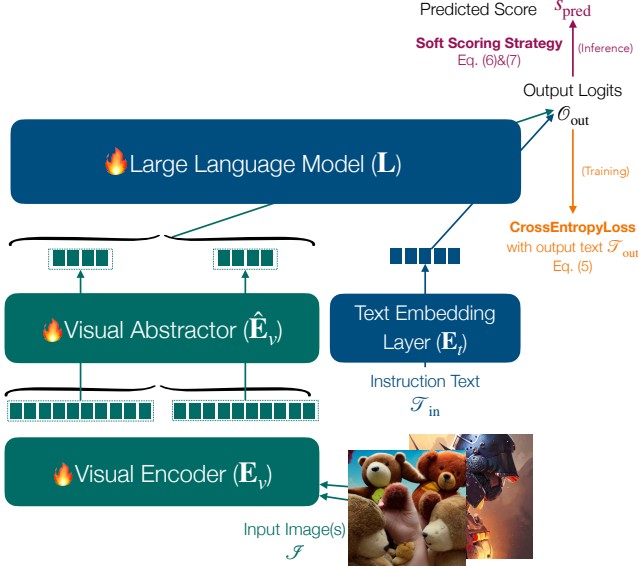

**Figure 4: The structure of LMM-based T2I-Scorer. We use CrossEntropyLoss as its training objective (*defined in Eq. 5*). During inference, we introduce the logit-based Soft Scoring Strategy (*defined in Eq. 6&7*) for quantitative evaluation.**

answers on quality-related questions but also provide quantitative evaluations. Afterwards, we further fine-tune the T2I-Scorer-IT with converted human opinion scores, into the **T2I-Scorer** (Sec. 4.2), which can provide more accurate multi-perspective T2I evaluation.

### 4.1 Stage 1: T2I-Scorer-IT

*4.1.1 Training Scheme.* We train the **T2I-Scorer-IT** under the general supervised fine-tuning (SFT) [31, 80] scheme for LMMs. Specifically, denote the instruction text as $\mathcal{T}_{\text{in}}$, the SFT loss $\mathcal{L}$ only supervises on the answer text $\{t_{\text{out}}^0, t_{\text{out}}^1, \ldots\}$ as follows:

$$O_{\text{out}}^{k-1} = \mathbf{M}(\mathcal{I}, \mathcal{T}_{\text{in}} \oplus \{t_{\text{out}}^0, \ldots, t_{\text{out}}^{k-1}\}) \text{ where } k \geq 1$$
$$\mathcal{L} = \sum_{k=0}^{K} \text{CrossEntropy}(O_{\text{out}}^{k-1}, t_{\text{out}}^k)/K \quad (5)$$

*i.e.* next token prediction loss on all tokens that are within the answer part of the conversation. Furthermore, as we have in total 397K question-answer pairs with very short total length, to best utilize training, we group up to three rounds of question answering for the same image(s) into one data item, resulting in **162K** data for the first training stage. With grouped data, the training time cost is reduced from 7 hours to 3 hours on 8*A100 GPUs.

*4.1.2 Soft Scoring Strategy for Quantitative Evaluation.* Primarily, the T2I-Scorer-IT is able to provide answers on open-vocabulary questions about T2I evaluation. Additionally, given that the answers can be converted to numerical levels (Eq. 1, 2, 3), the **T2I-Scorer-IT** can also predict quantitative scores for generated images.

For quantitative evaluation, a trivial strategy is to directly collect $f_{\text{YN}}(\mathcal{T}_{\text{out}})$ for an image with general *Yes-or-No* questions, *e.g.* '*Is the image with good quality?*', or $f_{\text{How}}(\mathcal{T}_{\text{out}})$ on general *How* questions, *e.g.* '*How is the image generated?*'. However, such scores can only

provide finite levels without enough precision. On the other hand, as mentioned in Sec. 3.2.1, as the answers in T2I-ITD are distinguishable from the first word, we can get the probabilities of each answer via softmax from the logits on first output token ($O_{\text{out}}^0$):

$$p(t_i) = \frac{e^{O_{\text{out}}^0(t_i)}}{\sum_{t_j \in C} e^{O_{\text{out}}^0(t_j)}} \tag{6}$$

where $C$ is the candidate answer set (*i.e.* {Yes,No} for *Yes-or-No* questions, {Good,Fair,Poor} for *How* questions). With the probabilities, we obtain $s_{\text{pred}}$ via the soft scoring strategy, as follows:

$$s_{\text{pred}} = \sum_{t_j \in C} p(t_j) f(t_j) \tag{7}$$

where $f$ is the general representation of the mappings in Eq. 1, 2, 3.

With the soft scoring strategy, the **T2I-Scorer-IT** is able to predict *real* quantitative scores for T2I generation, which is proven more accurate than the directly mapped scores as well as existing T2I metrics. It also shows high consistency between *Yes-or-No*-derived or *How*-derived scores, suggesting the effectiveness of the first stage training. We evaluate its quantitative ability in Sec. 5.4.

### 4.2 Stage 2: The T2I-Scorer

*4.2.1 Data Conversion from Human Opinion Scores.* For the second stage (fine-tuning), we would like to further train LMMs to provide subjective-aligned evaluations. Henceforth, we fine-tune the **T2I-Scorer-IT** with AIGIQA-20K [23] dataset. Following ITU [1] standards, we similarly convert the original scores (in range [0,5]) into 5-point likert scales: Bad,Poor,Fair,Good,Excellent.

The conversion is formulated as follows:

$$f_5 : \{\text{Bad,Poor,Fair,Good,Excellent}\} \to \{1, 2, 3, 4, 5\},$$
$$\text{where } f_5(\text{Bad}) = 1, f_5(\text{Poor}) = 2, f_5(\text{Fair}) = 3,$$
$$f_5(\text{Good}) = 4, f_5(\text{Excellent}) = 5, \tag{8}$$
$$L(s) = f_5^{-1}(\lceil s_{\text{gt}} \rceil)$$

where $s_{\text{gt}}$ is the original human opinion score. This conversion has allowed unified training objectives (as in Eq. 5) with the first training stage, and the same quantitative evaluation strategy (Eq. 7) as T2I-Scorer-IT. The training data template is defined as follows.

*4.2.2 Explicit Multi-task Learning.* Common T2I evaluation subjective studies [26, 67, 68] focus on two perspectives: (1) image-text alignment, *i.e.* how the generated image follows the given prompt, and (2) image quality, *i.e.* how is the perceptual quality of the generated images. To train the two perspectives under one model without mi, we instruct the models to answer explicit questions about the perspectives, as follows:

**Image-Text Alignment:** `<sys_hint> How does the image align with the text prompt?`
**Image Quality:** `<sys_hint> How is the picture quality of the image?`
where `<sys_hint>` defined the same as Sec. 3.2.1: `This is an image from the text-to-image generation, by the text prompt: <prompt>. Please reply with the simplest answer:`.

Similar as the first stage, we group the two perspectives into one multi-round question for each image in the training set. The second stage training only cost 13 minutes on 8*A100 GPUs.

## 5 EXPERIMENTS

### 5.1 Experimental Setups

We initialize the T2I-Scorer with the pre-trained checkpoint of mPLUG-Owl2 [70]. Before feeding to the LMM, images are first padded to square and then resized to $448 \times 448$. We use 8×NVIDIA A100 80G GPUs for training, with DeepSpeed ZeRO-3 optimization. We only use the final checkpoints for evaluation instead of picking checkpoints by validation set performance. All parameters of the LMM are updated during training.

### 5.2 Evaluation Datasets

We adopt the two most popular T2I evaluation datasets, AIGIQA-20K and AGIQA-3K, with fine-grained labels as evaluation datasets for T2I-Scorer-IT and T2I-Scorer. For AIGIQA-20K, 19K *(DALLE-2&DALLE-3 images excluded)* images are used for first stage training (*with* **T2I-ITD**), and 17K labeled images (further excluded Dream-Gaussian [10]) are used for fine-tuning with human opinion scores (stage 2). Henceforth, we evaluate **T2I-Scorer-IT** on all images (*i.e. blind to labels*) as well as the DALLE-2/DALLE-3 subset (*i.e. completely blind*); for the fine-tuned **T2I-Scorer**, we evaluate it on the 3K test set non-overlapped with the 17K training labeled images. Compare to normal random train-test splits, the split-by-generation-model strategy helps us to reach more reliable conclusions on how the metric can be applied to evaluate new T2I generation models in the future. For AGIQA-3K, neither its images nor its labels are used during training. We use it to evaluate the cross-set generalization ability of the proposed metrics. Following [54], we average the quality and alignment scores as an additional **overall** perspective, evaluated via human user study.

### 5.3 Baseline Methods

We include a wide variety of baseline models for comparison:

*General IQA Methods.* We include representative IQA methods in different categories as baseline models:

- **Statistical IQA Methods**: NIQE [37] and BRISQUE [36]. These methods are not trained on any IQA datasets.
- **Deep-learning-based IQA Methods**: including pure visual IQA methods NIMA [51], DBCNN [74] and MUSIQ [19], as well as CLIP-based IQA methods CLIP-IQA [53] and LIQE [76]. We compare **T2I-Scorer-IT** with their pre-trained models on general IQA datasets, and compare **T2I-Scorer** with their fine-tuned version on AIGIQA-20K [23] under the same train-test splits (17K *training set*, 3K *test set*, cross-model).

*Specialized T2I Metrics.* Despite comparison with general IQA methods, we further compare the **T2I-Scorer-IT** with two popular specialized T2I metrics: ImageReward [68] and HPS [67]. These two T2I metrics are trained with human preference data and *do not support* further fine-tuning with opinion scores, so we compare their official weights with **T2I-Scorer** on alignment perspective.

*Zero-shot LMMs.* To validate the improvements of the first tage, we further compare the **T2I-Scorer-IT** with representative zero-shot LMMs: LLaVA-v1.5-13B [30] and mPLUG-Owl2 [70] (*our base model*). Zero-shot LMMs are tested with their optimal settings [61].

**Table 2: Comparison of T2I-Scorer-IT with existing metrics. None of the methods are trained with human opinion scores on T2I-generated images. For the two *negative* question prompts (in \*), the scores are reversed for correlation calculation.**

| Dataset | | AIGIQA-20K (all) | | AIGIQA-20K (DALL-E) | | AGIQA-3K | |
|---|---|---|---|---|---|---|---|
| Type of Method | Method / *Question* | SRCC↑ | KRCC↑ | SRCC↑ | KRCC↑ | SRCC↑ | KRCC↑ |
| Statistical IQA Methods | NIQE [37] | 0.1436 | 0.0963 | 0.3196 | 0.2139 | 0.5329 | 0.3640 |
| | BRISQUE [36] | 0.3571 | 0.2424 | 0.0917 | 0.0630 | 0.4967 | 0.3648 |
| Pure Visual Deep IQA Methods | NIMA [51] | 0.5296 | 0.3640 | 0.5181 | 0.3559 | 0.6795 | 0.4856 |
| | DBCNN [74] | 0.5378 | 0.3736 | 0.7273 | 0.5055 | 0.6407 | 0.4428 |
| | MUSIQ [19] | 0.5287 | 0.3674 | 0.6938 | 0.4876 | 0.6297 | 0.478 |
| CLIP-based Deep IQA Methods | CLIP-IQA [53] | 0.3809 | 0.2610 | 0.5273 | 0.3648 | 0.6607 | 0.4656 |
| | LIQE [76] | 0.4926 | 0.3403 | 0.7062 | 0.4990 | 0.6972 | 0.4931 |
| Specialized T2I Metrics | ImageReward [68] | 0.5973 | 0.4230 | 0.4651 | 0.3169 | 0.6345 | 0.4516 |
| | HPS [67] | 0.6780 | 0.4912 | 0.6130 | 0.4302 | 0.6179 | 0.4371 |
| Zero-Shot LMMs | LLaVA-V1.5-13B [30] | 0.5995 | 0.4269 | 0.6254 | 0.4398 | 0.6723 | 0.4724 |
| *(under optimal settings [61])* | mPLUG-Owl2 [70] *(Our Base Model)* | 0.7019 | 0.5131 | 0.6674 | 0.4708 | 0.6481 | 0.4673 |
| **T2I-Scorer-IT (Ours)** | *Is the image with good quality?* | **0.8007** | **0.6045** | **0.8207** | **0.6130** | **0.8089** | **0.6098** |
| *(prompted with Yes-or-No Questions)* | *Is the image with poor quality?\** | 0.7323 | 0.5356 | 0.7400 | 0.5291 | 0.7920 | 0.5900 |
| | *Is the image generated well?* | 0.7863 | 0.5893 | **0.8060** | **0.5966** | 0.7819 | 0.5811 |
| | *Is the image generated poorly?\** | 0.7510 | 0.5559 | 0.7753 | 0.5643 | 0.7985 | 0.5970 |
| **T2I-Scorer-IT (Ours)** | *How is the quality of the image?* | **0.7985** | **0.6023** | **0.8084** | **0.5982** | **0.8021** | **0.6017** |
| *(prompted with How Questions)* | *How is the image generated?* | **0.7867** | **0.5906** | 0.7902 | 0.5825 | **0.8042** | **0.6036** |

## 5.4 Results of T2I-Scorer-IT

In Tab. 2, we compare the proposed **T2I-Scorer-IT** with different kinds of existing metrics for T2I evaluation. For the proposed **T2I-Scorer-IT**, it has reached state-of-the-art zero-shot performance on all three evaluation settings: it has more than **10%** improvement than any existing T2I metrics. Furthermore, we notice that existing models usually have some flaws: statistical IQA methods can bearly evaluate T2I generation; deep-learning-based IQA methods experience notable performance drop on many generative models (on **AIGIQA-20K** (all)), and the specific T2I metrics instead fall short on evaluations within a few models (on **AIGIQA-20K** (DALLE-3)). Consequently, these existing metrics might face challenges to simultaneously accurately *compare across models* and *evaluate on individual generated images*, while the proposed **T2I-Scorer-IT** can handle both scenarios better than any existing approaches.

Despite peer comparison, we further reach several important observations about the **T2I-Scorer-IT**: **1)** It shows similar accuracy on unseen images (DALLE-3 subset) in comparison to images *with pseudo labels during training*, proving the first training stage can learn general knowledge about T2I evaluation; **2)** Within *Yes-or-No* questions, *positive* prompting (*i.e.* good images receive Yes) in general shows **higher accuracy** than *negative* prompting, which may suggest the inductive bias from its training data that tends to ask questions in a positive manner. **3)** The performance of *Yes-or-No* questions and *How* questions are generally **on par**, showing that its evaluation ability is consistently elevated across question types.

Despite main results, we further qualitatively analyze pairwise evaluation ability of **T2I-Scorer-IT** in supplementary materials.

## 5.5 Results of T2I-Scorer

In this section, we evaluate the fine-tuned **T2I-Scorer** on multi-perspective T2I evaluation, as shown in Tab. 3 (**image quality**) and Tab. 4 (**image-text alignment**), as follows.

*Image Quality.* As shown in Tab. 3, the proposed **T2I-Scorer** is notably superior than existing IQA approaches on this setting. The

improvements are especially significant (leading all existing IQA methods by more than **10%**) on the test set of AIGIQA-20K, which contains only images generated by T2I models not included in the training set. This setting is meaningful as it measures whether the evaluator can robustly evaluate T2I-generated images in the future instead of over-fitting on the appearances of current T2I models, while the **T2I-Scorer** has proven its competitiveness on this meaningful setting. Additionally, **T2I-Scorer** also shows notable improvements upon base LMMs and the pre-traned stage 1 model (**T2I-Scorer-IT**) on both cross-model (**AIGIQA-20K** (test)) and cross-dataset (**AGIQA-3K**) evaluations, proving that our stage-2 fine-tuning is effective on image quality perspective.

**Table 3: Results of T2I-Scorer on Image Quality perspective, in comparison with existing fine-tuned IQA methods. We also include some zero-shot LMMs (*in italics*) into comparison to validate the effect of fine-tuning.**

| Dataset | AIGIQA-20K (test) | | AGIQA-3K | |
|---|---|---|---|---|
| Method | SRCC↑ | KRCC↑ | SRCC↑ | KRCC↑ |
| NIQE [37] *(zero-shot)* | 0.2614 | 0.1768 | 0.5329 | 0.3640 |
| BRISQUE [36] *(zero-shot)* | 0.2189 | 0.1493 | 0.4967 | 0.3648 |
| NIMA [51] | 0.7682 | 0.5728 | 0.7885 | 0.5910 |
| DBCNN [74] | 0.7589 | 0.5596 | 0.7107 | 0.5115 |
| CLIP-IQA+ [53] | 0.6102 | 0.4290 | 0.6869 | 0.4980 |
| LIQE [76] | 0.7984 | 0.6027 | 0.7583 | 0.5549 |
| *LLaVA-v1.5-13B [30]* | 0.5168 | 0.3607 | 0.6723 | 0.4724 |
| *mPLUG-Owl2 [70] (Base Model)* | 0.6107 | 0.4319 | 0.6481 | 0.4673 |
| *T2I-Scorer-IT (Stage 1 Model)* | 0.7367 | 0.5413 | 0.8021 | 0.6017 |
| **T2I-Scorer (Ours, Stage 2 Model)** | **0.8940** | **0.7174** | **0.8408** | **0.6525** |

*Image-Text Alignment.* Compared with the image quality perspective, the image-text alignment perspective marks a notably more difficult scenario: while **T2I-Scorer** is able to outperform all existing similarity-based T2I metrics as well as the baseline LMMs, none of the method achieves over 0.8 SRCC on this perspective, suggesting that there is still plenty of room for improvements to more

accurately evaluate the image-text alignment perspective of T2I generation. Among existing metrics, we notice that **T2I-Scorer** has more significant improvements than ImageReward on AIGIQA-20K test set, which has a notable longer average prompt length (**3 times** as long as AGIQA-3K), suggesting the proposed LMM-based metric can better understand more complex text prompts. As similar improvements are also observed on baseline LMMs in comparison with CLIP, this effect proves our aforementioned claim that LMMs may better understand complex prompts in T2I generation.

**Table 4: Results of T2I-Scorer on Image-Text Alignment perspective, in comparison with similarity-based metrics. All CLIP-based metrics can only allow ≤77 text tokens, so we trimmed the over-length prompts for them (labeled as [trim]).**

| Dataset | AIGIQA-20K (test) | | AGIQA-3K | |
|---|---|---|---|---|
| *Average Prompt Length* | **48.51** words | | 15.72 words | |
| Method | SRCC↑ | KRCC↑ | SRCC↑ | KRCC↑ |
| CLIP-RN50 [43][trim] | 0.2846 | 0.1924 | 0.5928 | 0.4204 |
| CLIP-ViT-B32 [43][trim] | 0.2814 | 0.1902 | 0.5770 | 0.4083 |
| CLIP-ViT-L14 [43][trim] | 0.2670 | 0.1809 | 0.5208 | 0.3618 |
| BLIP-2-ITM [29] | 0.3430 | 0.2340 | 0.5695 | 0.3991 |
| ImageReward [68] | 0.5625 | 0.3977 | 0.7298 | 0.5390 |
| HPS [67][trim] | 0.5729 | 0.4073 | 0.6349 | 0.4580 |
| LLaVA-v1.5-13B [30] | 0.3205 | 0.2185 | 0.6491 | 0.4633 |
| mPLUG-Owl2 [70] *(Base Model)* | 0.3528 | 0.2400 | 0.5885 | 0.4105 |
| *T2I-Scorer-IT (Stage 1 Model)* | 0.4799 | 0.3345 | 0.6765 | 0.4880 |
| **T2I-Scorer (Ours, Stage 2 Model)** | **0.6702** | **0.4888** | **0.7449** | **0.5512** |

## 5.6 Ablation Studies

*Effects of scaling up T2I-ITD.* In Fig. 5, we illustrate the accuracy change of **T2I-Scorer-IT** with different amount of **T2I-ITD** data used in first training stage. We notice that scaling up the **T2I-ITD** dataset consistently improve the accuracy of the **T2I-Scorer-IT**, which is not even saturated with the whole **T2I-ITD** used. The results have demonstrated that the pseudo-data training is not only useful, but also potentially further scalable to larger amount of data; on the other hand, the results also by-side validate that existing LMMs are still not sufficiently pre-trained for T2I evaluation.

*Effects of T2I-ITD on fine-tuned results.* While Tab. 2 has shown the significant direct improvement of the first stage training with **T2I-ITD** dataset, in Tab. 5, we further discuss its contributions to the fine-tuned results of **T2I-Scorer**. As shown in the table, the first stage training not only notably boosts the results on the **image quality** perspective as expected, but also slightly improves the **image-text alignment** perspective which is not the direct objective of the first stage training, which might be because prompts are included in the instruction template for the first stage training.

*Effects of explicit multi-task tuning.* In Tab. 5, we discuss the effects of the explicit multi-task fine-tuning scheme (as defined in Sec. 4.2), by comparing with the variant with implicit questions *How do you rate the quality of the image for dimension* i *(*i=1,2,3*)?* for the perspectives. As shown in the table, explicitly asking questions will notably improve performance for both perspectives. This suggests that the proposed explicit multi-task training can better

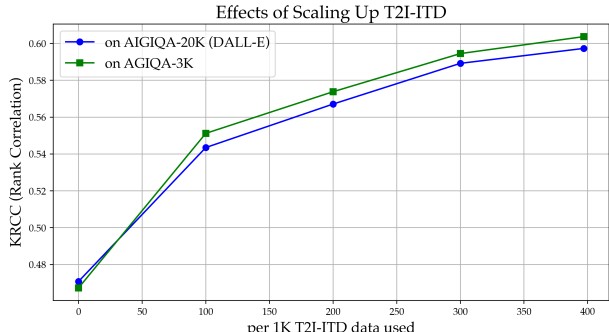

**Figure 5: Effects of scaling up the T2I-ITD dataset.**

inherit innate knowledge of LMMs and work as an effective scheme for multi-dimensional quantitative evaluation with LMMs.

**Table 5: Effects of the first stage training on T2I-ITD, on the final results of the second-sT2I-Scorer.**

| Image Quality | AIGIQA-20K (test) | | AGIQA-3K | |
|---|---|---|---|---|
| Variant / Metric | SRCC↑ | KRCC↑ | SRCC↑ | KRCC↑ |
| *w/o **T2I-ITD*** | 0.8495 | 0.6631 | 0.7987 | 0.5994 |
| *w/o Image Pairs in **T2I-ITD*** | 0.8802 | 0.6940 | 0.8248 | 0.6359 |
| **T2I-Scorer (Ours)** | **0.8940** | **0.7174** | **0.8408** | **0.6525** |
| **Image-Text Alignment** | **AIGIQA-20K** (test) | | **AGIQA-3K** | |
| Variant / Metric | SRCC↑ | KRCC↑ | SRCC↑ | KRCC↑ |
| *w/o **T2I-ITD*** | 0.6478 | 0.4614 | 0.7121 | 0.5207 |
| *w/o Image Pairs in **T2I-ITD*** | 0.6630 | 0.4735 | 0.7234 | 0.5341 |
| **T2I-Scorer (Ours)** | **0.6702** | **0.4888** | **0.7449** | **0.5512** |

**Table 6: Effects of Explicit Multi-task Learning (Sec. 4.2).**

| Image Quality | AIGIQA-20K (test) | | AGIQA-3K | |
|---|---|---|---|---|
| Variant / Metric | SRCC↑ | KRCC↑ | SRCC↑ | KRCC↑ |
| *w/o Explicit Multi-task Learning* | 0.8608 | 0.6795 | 0.8227 | 0.6310 |
| **T2I-Scorer (Ours)** | **0.8940** | **0.7174** | **0.8408** | **0.6525** |
| **Image-Text Alignment** | **AIGIQA-20K** (test) | | **AGIQA-3K** | |
| Variant / Metric | SRCC↑ | KRCC↑ | SRCC↑ | KRCC↑ |
| *w/o Explicit Multi-task Learning* | 0.6454 | 0.4589 | 0.7027 | 0.5114 |
| **T2I-Scorer (Ours)** | **0.6702** | **0.4888** | **0.7449** | **0.5512** |

## 6 CONCLUSION

In this work, we have proposed **T2I-Scorer**, the LMM-based evaluator for T2I (text-to-image) generation. It is trained by the **T2I-ITD** dataset, the GPT-4V-pseudo-labeled dataset with 397K question-answering pairs, and then further trained under an explicit multi-tasking training scheme to align with human-annotated opinion scores. The proposed **T2I-Scorer-IT** (pre-trained evaluator) and **T2I-Scorer** (fine-tuned evaluator) both achieve state-of-the-art accruaies under their respective settings. Furthermore, our evaluation is especially conducted on images across generation models or across different databases, demonstrating the generalized effectiveness of the proposed metric, and its eligibility to evaluate novel T2I generation models in the future. In the future works, we aim to explore how to further improve the image-text alignment ability on current evaluators, so as to fully unlock the strong text modeling capacity of LMMs for more holistic evaluation on T2I generation.

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
