# OpenReview forum: "T2I-Scorer: Quantitative Evaluation on Text-to-Image Generation via Fine-Tuned Large Multi-Modal Models"
_acmmm.org/ACMMM/2024/Conference — MM2024 Poster_

### Official Review · Reviewer_Uu9x · 2024-05-19

**Rating:** 3
**Confidence:** 3

**Summary:**

This paper introduces T2I-Scorer, a novel framework for evaluating the quality of text-to-image (T2I) generation models. It proposes a two-stage training methodology utilizing a large pseudo-labeled dataset (T2I-ITD) and fine-tuning to align with human judgment scores. The T2I-Scorer is designed to assess both image quality and image-text alignment, aiming to serve as a comprehensive metric for future T2I models.

**Strengths:**

- The two-stage training process is a creative solution to enhance the evaluation capabilities of large multi-modal models
- The use of a substantial pseudo-labeled dataset (397K question-answer pairs) provides a rich source for training the model.

**Limitations:**

- 397K question-answer pairs are generated by GPT-4v. How to ensure their quality of them, especially the reliability?
- The authors mention manual testing, and the 397K tests are believed to be time-consuming and costly, and the authors need to explain this
- The model's performance may heavily depend on the quality and representativeness of the pseudo-labeled data, which might not always be reliable or generalizable.
- The paper should provide more insight into how closely the model's judgments align with human perception and whether there are any noticeable discrepancies.

**Suitability:**

2

---

### Official Review · Reviewer_H17m · 2024-05-23

**Rating:** 4
**Confidence:** 3

**Summary:**

This paper constructs a large-scale VQA dataset, T2I-ITD, using GPT-4V. Utilizing this dataset, the authors trained T2I-Scorer-IT and T2I-Scorer in both zero-shot and fine-tune settings, achieving impressive performance on IQA tasks.

**Strengths:**

- The paper constructs a large-scale synthetic VQA dataset.
- In the zero-shot setting, the Soft Scoring Strategy effectively converts the model's output logits into scores, which makes sense to me.
- The metrics look promising, including both zero-shot and fine-tuning results.

**Limitations:**

I have some concerns about the **evaluation** of T2I-Scorer-IT. The T2I-ITD dataset consists of 20k GPT-4V synthetic images, with GPT-4V using the T2I model from DALL-E 3. T2I-Scorer-IT is trained on T2I-ITD, which means it uses images generated by DALL-E 3 during training. This might cause the metrics for T2I-Scorer-IT in Table 2 to **not be truly zero-shot**, since the AIGIQA-20K dataset includes images generated by DALL-E 3.

**Suitability:**

3

---

### Official Review · Reviewer_KVsA · 2024-05-26

**Rating:** 5
**Confidence:** 2

**Summary:**

This work addresses the problem of quantitatively benchmarking the output of T2I models. The authors propose T2I-Scorer, an LMM-based evaluator for T2I (text-to-image) generation. To conduct their analysis, the authors collect the T2I-ITD dataset, a GPT-4V-pseudo-labeled dataset with 397K question/answering pairs, and then perform a first fine-tuning, using a general supervised fine-tuning scheme, followed by a second alignment with human-annotated scores.

**Strengths:**

- I sympathize with the author's idea regarding the need of having more comprehensive quantitative benchmarks for T2I generative models
- I find this study well structured and easy to read
- I think this work could inspire further exploration in this direction

**Limitations:**

- I find that the author should have better clarified the differences between their approach and other scorers, based on LMMs
- I think that the presented Data Reliability study could have been done at a more significant scale, than having 1.5k samples given to 7 human experts
- The author could expand the supplementary material with many qualitative examples, to further prove their point

**Suitability:**

3

---

### Meta-Review · Area_Chair_uAzu · 2024-06-26

**Recommendation:** Accept (Poster)
**Confidence:** 5

**Metareview:**

This paper received mixed scores, including two "weak accept" and one "weak reject." Reviewer KVsA and Reviewer H17m were generally satisfied with the overall quality of the paper. After the rebuttal, Reviewer Uu9x lowered the score to "weak reject" but stated that they maintained the original score, without explaining the reason for continuing to insist on the negative score. Based on this, I recommend accepting this paper.